# In Vitro Antiviral Activity of Hyperbranched Poly-L-Lysine Modified by L-Arginine against Different SARS-CoV-2 Variants

**DOI:** 10.3390/nano13243090

**Published:** 2023-12-06

**Authors:** Federico Fiori, Franca Lucia Cossu, Federica Salis, Davide Carboni, Luigi Stagi, Davide De Forni, Barbara Poddesu, Luca Malfatti, Abbas Khalel, Andrea Salis, Maria Francesca Casula, Roberto Anedda, Franco Lori, Plinio Innocenzi

**Affiliations:** 1Laboratory of Materials Science and Nanotechnology (LMNT), CR-INSTM, Department of Biomedical Sciences, University of Sassari, Viale San Pietro 43/B, 07100 Sassari, Italy; federico.fiori@studenti.unipg.it (F.F.); cossufrancalucia@libero.it (F.L.C.); f.salis3@studenti.uniss.it (F.S.); dcarboni@uniss.it (D.C.); lstagi@uniss.it (L.S.); luca.malfatti@uniss.it (L.M.); 2ViroStatics srl, Viale Umberto I, 46, 07100 Sassari, Italy; d.deforni@virostatics.com (D.D.F.); b.poddesu@virostatics.com (B.P.); f.lori@virostatics.com (F.L.); 3Department of Chemistry, College of Science, United Arab Emirates University, Al Ain P.O. Box 15551, United Arab Emirates; abbask@uaeu.ac.ae; 4Department of Chemical and Geolocial Sciences, University of Cagliari, Cittadella Universitaria SS 554 Bivio Sestu, 09042 Monserrato, Italy; asalis@unica.it; 5Department of Mechanical, Chemical and Materials Engineering, University of Cagliari, Via Marengo, 2, 09123 Cagliari, Italy; 6Porto Conte Ricerche srl, Strada Provinciale S.P. 55, Loc. Tramariglio, 07041 Alghero, Italy; anedda@portocontericerche.it

**Keywords:** L-lysine, L-arginine, hyperbranched polymers, antiviral, SARS-CoV-2

## Abstract

The emergence of SARS-CoV-2 variants requires close monitoring to prevent the reoccurrence of a new pandemic in the near future. The Omicron variant, in particular, is one of the fastest-spreading viruses, showing a high ability to infect people and evade neutralization by antibodies elicited upon infection or vaccination. Therefore, the search for broad-spectrum antivirals that can inhibit the infectious capacity of SARS-CoV-2 is still the focus of intense research. In the present work, hyperbranched poly-L-lysine nanopolymers, which have shown an excellent ability to block the original strain of SARS-CoV-2 infection, were modified with L-arginine. A thermal reaction at 240 °C catalyzed by boric acid yielded Lys-Arg hyperbranched nanopolymers. The ability of these nanopolymers to inhibit viral replication were assessed for the original, Delta, and Omicron strains of SARS-CoV-2 together with their cytotoxicity. A reliable indication of the safety profile and effectiveness of the various polymeric compositions in inhibiting or suppressing viral infection was obtained by the evaluation of the therapeutic index in an in vitro prevention model. The hyperbranched L-arginine-modified nanopolymers exhibited a twelve-fold greater therapeutic index when tested with the original strain. The nanopolymers could also effectively limit the replication of the Omicron strain in a cell culture.

## 1. Introduction

The frequent mutations of SARS-CoV-2 that have generated many different variants [1,2], including the Omicron variant [3,4], have shown increasing infectivity and transmissibility. Mutations on the receptor-binding domain (RBD) of the spike protein (S) cause a change in the surface charge of the virus, which may lead to an enhanced capacity to attract the human angiotensin-converting enzyme 2 (hACE2) [5,6]. The spike protein, a glycosylated trimer, emerging from the envelope of SARS-CoV-2 mediates the entry into the host cell via the identification of the receptor and membrane fusion [7]. An important role in infecting the target cell is played by the infiltration of the S protein to cellular heparan sulphate (HS), which is negatively charged, and a sulphated polysaccharide molecule [8]. Cellular HS is a necessary co-factor for SARS-CoV-2 infection by interacting with the receptor-binding domain of the spike glycoprotein that is shifted to an open conformation, facilitating the hACE2 binding [9]. The invasion of the target cells, in fact, begins with the attachment of the spike protein S to HS that can be bound in a length- and sequence-dependent way [7]. Negatively charged heparan sulfate proteoglycans (HSPGs) have a strong affinity with positively charged amino acid residues (Arg346, Arg355, Lys444, Arg466, and possibly Arg509) in the receptor-binding motif (RBM) [8]. The electrostatic interaction between HS and the viral spike promotes the conformational transition in the RBD from an inactive (closed) state to an active (open) state. This conformational change favours the simultaneous RBD-hACE2 binding. Besides the five charged amino acid residues, possibly six other RBD amino acids (Phe347, Ser349, Asn354, Gly447, Tyr449, and Tyr451) are involved in the formation of H-bonding with HSPGs that stabilize the binding.

It has been observed that the mutations in S alter the surface charge that favours virus invasiveness and immune escape. A comparative analysis of the SARS-CoV-2 variants has revealed a tendency to increase the total positive charge in the spike protein in the RBM. The study has shown an evolution of the electrostatic charge, with the early variants (Alpha, Beta, Epsilon, and Lamba) gaining a +e charge. Subsequent variants (Gamma, Delta, Eta, Iota, Kappa, and Mu) gained +2e, while Omicron, in turn, gained a much higher charge, +5e [4]. An increase in the electrostatic potential surface and polarity of the spike protein is predicted to increase its affinity for the negatively charged HPSG and the negatively charged receptor hACE2. This change gives Omicron a higher infectivity; one of the reasons for Omicron’s immune-escaping capability is its increased positive surface change and higher binding affinity to hACE2 [4]. Several theoretical calculations have been performed to model the spike interactions with the hACE2 mediated by HS as a function of the electrostatic potential at the interface. This is an adaptive virus variation to enhance its binding capability to hACE2. The S surface charge model underlines that the spike charge plays a primary role even if the real case is much more complex. In fact, ions mediate the protein electrostatic surfaces in physiological conditions, and the effective interactions in a complex fluid are difficult to model.

In previous work, we used poly-L-lysine hyperbranched nanoparticles as an antiviral system to inhibit the replication of the SARS-CoV-2 original strain [10]. The rationale behind the synthesis was the use of an amino acid as a precursor, which is non-toxic and has demonstrated inhibitory effects in the replication of several RNA and DNA viruses [11]. The nanopolymer dimension was adjusted to match that of the virus, while the surface charge was positive and governed by L-lysine residues. The experimental data showed that the nanopolymer can prevent virus entry into the cells. In the present work we extended the use of the nanopolymer to inhibit the viral replication of different SARS-CoV-2 variants, namely Delta and Omicron, in a prevention model. We modified the composition of the hyperbranched poly-L-lysine nanopolymers by co-reacting L-lysine with another amino, acid, L-arginine (see Figure 1). L-arginine is an amino acid characterized by the presence of a guanidinium group appended to an amino acid framework. At physiological pH, the carboxylic acid is deprotonated (−COO^−^) and both the amino and guanidinium groups are protonated, resulting in a bivalent cation. The addition of L-arginine is expected to modify the surface charge, changing the interaction with the virus and cellular receptors. As we have seen, the charge is critical in designing antivirals active against SARS-CoV-2 variants. An increase in the positive surface charge can enhance the binding affinity to the cellular hACE2, inhibiting or hampering the docking capability of the virus. Among the possible strategies, two can be used to reduce the capability of the virus to infect the cell: the first option is using charged nanoparticles that are able to bind electrostatically to the virus, thus acting as a kind of antibody; the second strategy would be to use nanoparticles with a highly positive charge capable of competing with the virus for binding to the cellular receptor, thus impeding the virus in entering the cell. The surface design of antiviral nanosystems is therefore critical to modifying and adapting their properties according to mutations in the virus surface charge.

## 2. Materials and Methods

L-lysine (Lys; L5501) ((S)-2,6-diaminocaproic acid) powder (crystallized, ≥98.0% (NT), H_2_N(CH_2_)_4_CH(NH_2_)CO_2_H) and L-arginine (Arg; W381918) ((S)-2-Amino-5-guanidinopentanoic acid) powder (crystallized, 99%, (H_2_N)(HN)CN(H)(CH_2_)_3_CH(NH_2_)CO_2_H), were purchased from Sigma Aldrich, St Louis, MO, USA. Poly-L-lysine hydrochloride (P2658, mol wt 15,000–30,000) was purchased from Sigma Aldrich (St Louis, MO, USA) and used as a reference. Boric acid (H_3_BO_3_; 402766), (≥99.5%) was purchased from Carlo Erba Reagents, Rodano (Rodano, MI, USA), ITA. All chemicals were used without further purification. Milli-Q water was used for synthesis and analysis. The purification steps were carried out using benzoylated membranes from dialysis tubing (avg. flat width 32 mm, 1.27 in., molecular weight cut-off = 2000 Da, Sigma Aldrich, St Louis, MO, USA). 

### 2.1. Nanopolymers Preparation

All nanopolymers were synthesized by a thermal polymerization method according to the procedure reported in our previous works [10,12] using 1 g of the amino acid of interest and increasing the molar ratios of L-arginine while keeping the amount of boric acid (BA) also fixed. Hyperbranched poly-L-lysine (HBPL) was replicated by thermally polymerizing L-lysine catalyzed by boric acid. Five different samples were prepared using the following ratios: Lys:BA:Arg = 1:1:0.1 (LBA0.1); Lys:BA:Arg = 1:1:0.25 (LBA0.25); Lys:BA:Arg = 1:1:0.5 (LBA0.5); Lys:BA:Arg = 1:1:1 (LBA1), and Lys:BA:Arg = 1:1:2 (LBA2). Two reference samples were also synthesized, one without L-lysine: BA:Arg = 1:1 (BAA) and the other without L-arginine (sample HBPL): Lys:BA = 1:1. The precursors were mixed in a mortar, placed in a ceramic crucible, and then heated up to 240 °C for 5 h in air. The mixture was cooled down to 20 °C before any further treatment. The obtained brown-black solid powder was dispersed in Milli-Q water, sonicated for 1 h, and then centrifugated at 9000 rpm for 30 min. The supernatant was collected and dialyzed using Milli-Q water for 24 h, replacing the water every 12 h. Then, the resulting nanomaterials were freeze-dried for 24 h with a Lio 5P device and kept at 4 °C before characterization.

The pH of the precursor solutions was measured at a concentration of 1 mg mL^−1^ for each sample (precursors and nanopolymers) using a bench pH meter (XS PH 80+ DHS) (see Appendix A). The solutions were prepared using Milli-Q water (pH = 6.8). All measurements were taken in triplicate.

### 2.2. Materials Characterization

Thermogravimetric analysis (TGA) and differential scanning calorimetry (DSC) analysis were performed using an SDT Q600 device (TA instruments). All the TGA-DSC analyses were carried out under an inert N_2_ atmosphere (flow rate of 20 mL min^−1^) with a ramp of 10 °C min^−1^ up to 400 °C.

Fourier-transform infrared (FTIR) analysis was carried out using an infrared Vertex 70 interferometer (Bruker). The FTIR absorption spectra were recorded in the 4000–400 cm^−1^ range with a 4 cm^−1^ resolution and 128 scans. The spectra were acquired using KBr pellets (sample: KBr = 1:500).

All NMR spectra were acquired using a Bruker Advance instrument at a 600 MHz proton frequency (Bruker BioSpin GmbH, Karlsruhe, Germany). A Bruker BBI 5 mm probe with z-gradients was used. All measurements were performed at T = 298 K (Bruker BVT3000 and BCU05 temperature control units). One-dimensional ^1^H-NMR and two-dimensional ^1^H-^13^C HSQC and ^1^H-^1^H COSY measurements were acquired using JCH = 145 Hz and long-range coupling optimized sequences, respectively. Approximately 10 mg of each sample was weighed and dissolved in 1 mL of a 50 mM phosphate buffer solution in D_2_O (99.9%, Cambridge Isotope Laboratories Inc., Andover, MA, USA) at pH 4.5 with added trimethylsilylpropanoic acid (TMSP) used as a chemical shift standard (δ = 0 ppm).

Atomic force microscopy (AFM) analysis was performed with an NT-MDT Ntegra microscope (Moscow, Russia) at a 0.5 Hz scan speed in the semicontact mode using a silicon tip with typical resonant frequency of 240 kHz and a typical force constant of 11.8 N/m.

Transmission electron microscopy investigation was performed using a JEM 1400 Plus TEM (Jeol Ltd., Tokyo, Japan) operating at 80 kV. Prior to the observations, the sample dispersions were sonicated, dropped on a carbon-coated 200-mesh copper grid, and dried at 50 °C. 

The zeta potential (ζ) and hydrodynamic diameter (size) of the nanoparticles dispersed in aqueous solutions were measured by dynamic light scattering (DLS) using a Zetasizer Nano ZSP device (Malvern Instruments, Westborough, MA, USA) in the backscatter configuration (θ = 173°; laser wavelength of λ = 633 nm). The scattering cell temperature was fixed at 298 K, and the data were analyzed through the Zetasizer software version 7.03. The samples were prepared by dissolving solid samples in Milli-Q water (1 mg mL^−1^). Samples were left under rotation for one hour at 25 °C before analysis. Each measure was the average of 3 runs of 14 measures each and was replicated three times. The final value was calculated by taking the average of the three replications.

### 2.3. Viral Isolate

The human 2019-nCoV strain 2019-nCoV/Italy-INMI1 was isolated in Italy (ex-China) from a sample collected on 29 January 2020 from Istituto Lazzaro Spallanzani, Rome, Italy [13]. Delta variant B.1.167.2 and Omicron variant BA.1 viral strains were kindly provided by the Department of Molecular and Translational Medicine, Section of Microbiology and Virology, University of Brescia Medical School, Brescia, Italy [14,15,16,17].

### 2.4. Cytotoxicity

The cytotoxicity (i.e., reduction in cell viability) of the nanomaterials was determined by a standard MTS (3-(4,5-dimethylthiazol-2-yl)-5-(3-carboxymethoxyphenyl)-2-(4-sulfophenyl)-2H-tetrazolium) assay in the Vero E6 cell line (renal epithelial cells of green vervet, ATCC CRL-1586) and microscopic observation of the integrity of the cell monolayer, in duplicate. Vero E6 cells were kept in culture at their optimal density based on the indications reported by the ATCC. On day 1 of the experiment, the cells were transferred to 96-well plates (10,000 cells per well). On day 2 of the experiment, the cells were treated with different concentrations of each compound. On day 5, the cell viability was determined according to the described methods. A value of TD_50_ (toxic dose 50, the dose that reduces cell viability by 50%) was calculated. The compound remdesivir (virus RNA polymerase inhibitor antiviral drug) was used as a control.

### 2.5. Antiviral Activity

Efficacy against SARS-CoV-2 replication in suspension was measured in a viral replication assay in prevention and post-infection models. Vero E6 cells were maintained in culture at their optimal density according to indications reported by the ATCC. In the prevention model, on day 1 of the experiment, the cells were transferred to 96-well plates (10,000 cells per well). On day 2 of the experiment, the cells were first treated with different concentrations of each nanopolymer for 1 h and then infected with the SARS-CoV-2 virus (multiplicity of infection of 0.01). On day 5 (72 h after infection), the antiviral activity of the compounds was established by determining the viral replication with an ELISA test (SARS-CoV-2 Nucleo-capsid Detection ELISA Kit, Sino Biological, Beijing, China) and confirmed through microscopic observation of the protection from the cytopathic effect of the virus on the cell monolayer. Each culture condition was analyzed in duplicate for each sample. An IC_50_ value was therefore calculated where possible (inhibitory concentration 50, the dose that inhibits virus replication by 50%). The compound remdesivir was used as a control. 

In the post-infection model, on day 1 of the experiment, the cells were transferred to 96-well plates (10,000 cells per well). On day 2 of the experiment, the cells were infected with the SARS-CoV-2 virus (multiplicity of infection of 0.01) and then treated with different concentrations of the nanopolymers for 1 h. On day 5 (72 h after infection), the antiviral activity of compounds was established by assessing the viral replication with an ELISA test (SARS-CoV-2 Nucleo-capsid Detection ELISA Kit, Sino Biological). The antiviral activity was confirmed through microscopic observation of the protection from the cytopathic effect of the virus on the cell monolayer. Each culture condition was analyzed in duplicate for each sample.

## 3. Results and Discussion

The syntheses of the nanopolymers were performed by thermally co-polymerizing L-lysine and L-arginine in various ratios using boric acid to catalyze the amidation reaction.

### 3.1. Hyperbranched Nanopolymer Characterizations

TGA-DSC was used to evaluate the compounds’ thermal stability under the reaction conditions used for the syntheses. The analyses additionally provided important information on how temperature affects amino acid reactions. A comparison of the thermal analyses of L-lysine and L-arginine showed that the latter was thermally more stable. On the other hand, L-lysine was among the amino acids most susceptible to thermally induced degradation (Figure 2).

At 400 °C, L-lysine and L-arginine lost 80% and 55% of their weight, respectively. L-arginine showed two endothermal effects, the first at 219 °C without any mass loss and the second at 236 °C with a 30% mass loss. The first peak of heat flow in Figure 2a can be attributed to a molecular rearrangement of the guanidinium group. The second, formally involving the loss of 1 mol of NH_3_ and 1 mol of water, can be assigned to a double intramolecular cyclization. The first cycle was obtained through the internal nitrogen of the guanidinium group attacking the α-carbon and expelling NH_3_, leading to a proline derivative. At this point, the primary amino group of the guanidinium residue causes a nucleophilic reaction that attacks the carboxylic group. The reaction produces a cyclic amide, a lactam, with a loss of 1 mol of water, resulting in a two-ring structure obtained through the combination of creatinine and proline moieties [18]. 

The TGA of boric acid exhibited two major endothermic signals that peaked at 130 and 160 °C, respectively, due to dehydration and the production of metaboric acid (HBO_2_). The first-derivative curves of the TGA revealed that the endothermic events corresponded to the weight loss of the samples. The classic peptide reaction occurred through the condensation of carboxylic groups and amines. The DSC curves showed that the endothermic amidation reaction occurred at the same temperature in the L-lysine and L-arginine at 235 °C.

When boric acid and L-arginine were combined, the reactions occurred at slightly higher temperatures, with the amidation reaction peaking at 242 °C with a small shoulder at 223 °C. Noticeably, with the addition of boric acid, the first endothermic event, peaking at 223 °C, was connected with a weight loss, as indicated by a shoulder in the first-derivative curve, which did not occur in pure L-arginine. This loss can be explained by the interaction between boric acid and L-arginine, which significantly impacted the thermal reaction pathway. While bare L-arginine lost water and ammonia to give an intramolecular cyclic amidation, in combination with boric acid, it formed a cyclic catalytic intermediate instead [19]. The second endothermic event can thus be related to the formation of the amidation product, in which the L-arginine moiety, activated by the boric acid, reacted with a second residue of L-arginine or L-lysine, affording nanopolymers, whose nature depended upon the Lys:Arg ratio. The same behavior can also be envisaged for the amidation of L-lysine catalyzed by boric acid (see Appendix A). While the DSC profile of the pure L-lysine showed a single peak at 236 °C, that of the equimolar mixture of L-lysine and boric acid revealed two close endothermic events, peaking at 196 and 214 °C, that can be related to the formation of the boric-acid–lysine intermediate and the resulting amidation product, respectively. This behavior was observed in all the samples prepared with different Lys:Arg ratios catalyzed by boric acid (see Appendix A). In particular, the samples with ratios equal to 1:0.1 showed a profile similar to that of BA:Lys = 1:1. This is not surprising, since the amount of L-arginine was only 10%. Starting from the 1:0.25 ratio, the first endothermic event was observed in the curve as a weak shoulder. At the same time, the L-arginine amount increased, giving rise to a single band convoluting the two events: the formation of the intermediate and subsequent amidation. The shape of the DSC curve can be explained by considering that, in all the samples, the amount of boric acid, which is a limiting factor for forming the cyclic intermediates, was kept constant. In contrast, the amount of amino-acid moieties progressively rose to a 1:3 ratio in the sample with Lys:Arg = 1:2. This tendency appeared more evident when examining the first-derivative profile of the weight loss (see Appendix A), where the two peaks related to the weight loss gradually transformed into a single band with a shoulder.

Figure 3a shows the FTIR absorption spectra of L-lysine and L-arginine in the 1800–1500 cm^−1^ range (the full spectra are reported in Appendix A). The spectra were collected using the as-purchased amino acid powders. The absorption spectrum of L-lysine was characterized by a main band peaking at 1581 cm^−1^ assigned to -COO^-^ antisymmetric stretching, a shoulder at 1615 cm^−1^ due to the antisymmetric stretching of -NH_3_^+^, and the rocking band of the same group at 1515 cm^−1^. The L-arginine spectrum showed four absorption bands at 1722 cm^−1^ (ν_s_ C=O), 1728 cm^−1^ (bending NH_2_), 1623 cm^−1^ (ν_s_ C=N in the guanidinium group), and 1555 cm^−1^ (bending N-H). The observation of these bands upon reaction can give some clues about the changes in the structure (Figure 3b). The reaction catalyzed by boric acid induced polymerization, as shown by the rise in the amide I band around 1650 cm^−1^. Previous works have indicated that a reaction between L-lysine and boric acid forms a hyperbranched polymer [10,12], HBPL (Figure 3b). The addition of a small amount of L-arginine, sample LBA0.1, did not change the FTIR spectrum for the reasons we have explained before (vide supra). At higher L-arginine concentrations, the spectra showed a distinct amino acid signature that overlapped the HBPL spectrum, indicating that not all the amino acid functional groups reacted. The full FTIR absorption spectra of the Lys-Arg samples are reported in Appendix A.

Figure 4 shows the NMR spectra in the 0.4–4.6 ppm range of the samples prepared with increasing amounts of L-arginine. The reference spectra of pure L-lysine and L-arginine are reported in Appendix A. Significant NMR signal broadening suggests the formation of slowly reorienting macromolecular structures. The intensity of the signals at 1.4 and 3.0 ppm, assigned to the γ- and ε-CH_2_ protons of L-lysine, respectively, decreased with increasing the amount of L-arginine in the mixture. The signal from ε-CH_2_ protons, also observed in commercial poly-L-lysine (Appendix A), showed a downfield shift as a function of increasing amounts of L-arginine, suggesting a progressive amide bond formation until the Lys:Arg ratio reached the 1:1 value. The sample with the lowest content of L-arginine (LBA0.1) showed a signal at 3.75 ppm that progressively shifted to 3.91 ppm in LBA0.25 and to 3.96 ppm in LBA0.5 and LBA1 (See Appendix A). This trend suggests that the signal was due to the α-CH_2_ groups next to the amide bonds of L-lysine. In particular, the rise in the signals at 4.15 and 4.25 ppm indicates the formation of a hyperbranched polymeric structure via amide bonding, which was in agreement with FTIR analysis and previously published data [20,21]. 

The signal at 4.38 ppm corresponded to α-CH protons in the dendritic structure formed by the reaction of the carboxyl group with the secondary amino group of the guanidinium star moiety in L-arginine. This signal also rose in intensity with the increase in the amount of arginine, confirming this possible attribution. The NMR data suggest that increasing the L-arginine content in the mixture caused the formation of a more branched and less linear structure. 

With the increase in the amount of L-arginine co-reacted with L-lysine, the NMR spectra in the same range were dominated by signals that were due to polymerized L-arginine, as can be observed by comparing the spectra with that of pure L-arginine. Furthermore, the NMR data suggest that the reaction of L-lysine with L-arginine at low concentrations formed an interconnected structure. L-arginine became, in this case, a modifier of the main L-lysine hyperbranched polymeric structure. On the other hand, at high concentrations, L-arginine formed its own interconnected network, and the polymer was intrinsically more disordered with fewer linear structures. 

Figure 5 shows a two-dimensional NMR spectrum that displays the homonuclear (^1^H-^1^H COSY) correlations for the sample with the lowest concentration of L-arginine (LBA0.1), confirming the previous assumptions. In particular, the signals of the α-CH proton and their β-CH_2_ siblings (3.81 and 1.88 ppm, respectively), usually correlated with peptide formation, were shifted downfield (4.23 and 1.94 ppm), indicating the formation of hyperbranched polymers (HBP), as previously reported for hyperbranched poly-L-lysine (HBPL) [10]. The formation of a hyperbranched structure is not surprising since the Lys:Arg molar ratio was only 1:0.1, and, thus, adding a small amount of L-arginine did not significantly alter the hyperbranched nature of the nanopolymer. This hyperbranched character was also confirmed by another two-dimensional NMR spectrum reporting a proton–carbon heteronuclear correlation (^1^H-^13^C HSQC) (see Appendix A). Here, the signals of the α-C carbon and the correlated α-CH proton of the amino acids involved in the peptide chains were downshifted at 56.44 (α-C carbon) and 4.22 ppm (α-CH proton), respectively, for the L-lysine moiety and to 56.23 and 4.28 ppm for L-arginine.

The thermal polymerization of L-lysine and L-arginine led to the formation of nanopolymers with branched peptidic chains. The protein-like nature was governed by the pH-dependent protonation states of the L-lysine- and L-arginine-derived samples [20]. As shown in Figure 6, the two amino acids could assume four protonation–deprotonation states with different charges and pKa values.

The thermally induced amidation gave rise to polypeptide nanopolymers bearing amino acid side-chains that control the overall surface charge of the nanoparticle and thus have an important effect in its interaction with viruses. In particular, Figure 6 suggests that the amino acid side-chains, at physiological pHs, should be mostly in the +1 states, Lys^+^ and Arg^+^, respectively.

The sizes of the different nanoparticles were assessed by TEM (Figure 7). The analysis of the samples at various lysine/arginine ratios showed that the nanopolymers were made of a wide range of structures featuring different electronic densities, which were related to different polymerization degrees. The smaller structures had an average size in the range of 20–50 nm and appeared to be more polymerized with respect to the larger particles, which showed a size ranging from 80 to 200 nm. AFM (see Appendix A) confirmed the presence of harder nanoparticles with a size in the range of 40–60 nm, which appeared embedded into a softer structure, compatible with that of not completely polymerized particles. 

DLS was therefore used to evaluate the stability in solution of the nanopolymers (Figure 8). Regardless of the complex morphology revealed by TEM, the hydrodynamic radius, as measured by DLS, was in the range of hundreds of nanometres, indicating that the smaller nanoparticles tended to form aggregates when dissolved in water. In more detail, the hydrodynamic radius of the nanopolymers was in the range between 150 and 300, similar to the average size of SARS-CoV-2 virions, i.e., 60–140 nm [22,23], except for sample LBA2, which was out of the range of interest.

The absolute ζ potential value indicates the relative stability of nanoparticles. A positive ζ potential generally indicates that the surface of a particle has a positive charge and is a measure of the electrostatic repulsion between particles or surfaces in a liquid medium. BAA and LBA2 showed the lowest ζ potential (Appendix A), while the other particles had a positive ζ potential close to +20 mV. The particles with a smaller ζ potential tended to aggregate or precipitate. This was observed in the particles with the highest content of L-arginine, which makes them not suitable for biological applications.

It is important to stress that the ζ potential is not the only factor that determines the surface charge of a particle or surface. The measurement can also be affected by different variables, such as the pH, the ionic strength, and the presence of charged molecules or ions in the surrounding fluid. Therefore, a direct experimental evaluation of the surface charge and surface state is not simple. In the present case, the values obtained when measuring the ζ potential were not taken in a cell culture medium but in an aqueous solution, which does not represent the biological medium where a viral infection occurs. This choice was dictated by the need for avoiding interference due to the heterogeneity of the medium that could have affected the measurements. Also, in the case of SARS-CoV-2 strains, the experimental assessment of the surface properties is complicated. Therefore, only theoretical calculations based on the amino acid residues in the spike are generally reported in the literature. Theoretical calculations, in fact, do not take into account the role of water and ions in bridging the interaction between the virus spike and the cell receptors.

### 3.2. Cytotoxicity

A standard MTS cytotoxicity assay was used to assess the cytotoxicity of the different compositions of the nanopolymers. The test uses soluble tetrazolium salts, such as 3-(4,5-dimethylthiazol-2-yl)-5-(3-carboxymethoxyphenyl)-2-(4-sulfophenyl)-2H-tetrazolium, indicated as MTS, which are reduced by cellular nicotinamide adenine dinucleotide (phosphate)-dependent oxidoreductase enzymes in the presence of an intermediate electron acceptor (phenazine ethosulfate). The formazan derivative that is produced is quantified by spectrophotometry and indicates the metabolic activity and, thus, the viability of cells. In the present case, we used the standard test dedicated to nanomaterials [24] using an MTS assay as an in vitro cytotoxicity assay in accordance with previous reports [25]. Vero E6 cells were exposed to increasing nanopolymer concentrations, reaching the 50% cytotoxic concentration (CC50) value (see Figure 9). Remdesivir was used as a reference to compare the present results (see Appendix A) [26]. The CC50 was higher than 100 μg mL^−1^ for all the nanopolymers except for LBA1 and LBA0.5, whilst that of Remdesivir was higher than 6 μg mL^−1^. The compounds with higher amounts of L-arginine seemed to have a stronger effect on cell survival. This might have been caused by a putative higher number of proline moieties in the structure of the polymers, which were formed, as said before, during the thermal treatment. Such structural factors have been associated with a cytotoxic effect through protein translation inhibition and other still-unknown mechanisms that might be involved [27]. The value of 500 μg mL^−1^ represents the highest dose that could be tested because of the DMSO solvent concentration limits in cell culture.

### 3.3. Antiviral Activity

An antiviral assay was performed in parallel with the cytotoxicity experiments, in a prevention and post infection model, using the SARS-CoV-2-permissive cell line Vero E6 from the same culture. In the prevention model, the cells were first seeded into 96-well plates and exposed for 1 h to different nanopolymer concentrations. Afterwards, the cells were infected with the SARS-CoV-2 original, Delta, and Omicron strains (with a multiplicity of infection, m.o.i., of 0.01) and finally cultured for 72 h.

In the post-infection model, on the contrary, the cells were first cultured with the SARS-CoV-2 strains for 72 h and then exposed to increasing nanopolymer concentrations for 1 h. In both models, the viral replication was assessed by ELISA to quantify the SARS-CoV-2 nucleoprotein. 

The antiviral efficacy data (i.e., SARS-CoV-2 nucleocapsid protein expressed as % of control, mean, and standard deviation) obtained in the ELISA assay for all the nanopolymers are shown in Figure 10 for the different viral strains, i.e., original, Delta, and Omicron (a, b, and c panels refer to the prevention model, while c, d, and e refer to the post-infection model). 

The nanopolymers were highly effective against the original strain, and at the greatest concentration, they could entirely block viral propagation. The data obtained from the prevention model show that all the L-arginine-modified nanopolymers had a stronger antiviral activity compared to HBPL at concentrations equal to or higher than 20 μg mL^−1^, except for LBA0.5, which still overcame the HBPL performance at a concentration of 100 μg mL^−1^. This result indicates that the co-polymerization with arginine moieties is an effective method to enhance the antiviral effect. In the case of the Delta virus, the antiviral activity was low and became visible only at 100 μg mL^−1^, with HBPL being the most effective. The lack of a linear trend in the antiviral efficacy as a function of the L-lysine/L-arginine ratio can be explained by considering that the ELISA results depended on both the antiviral properties of the nanoparticles and their cytotoxicity. As an extreme case, by using an ELISA essay, a theoretical nanoparticle concentration achieving a 0% cell viability produced the same effect as a nanoparticle concentration that is completely effective against virus replication. Indeed, both were capable of minimizing the amount of SARS-CoV-2 nucleoprotein, reducing the probability of viral replication to zero. In intermediate cases, as with the LBA nanopolymers, both the reduction in cell viability (cytotoxicity) and antiviral efficacy contributed to the ELISA, producing a non-linear trend in the data sets.

The nanopolymers were generally highly active towards the Omicron strain, and the effect appeared to be correlated with the L-arginine content, with HBPL being the least effective. These results suggest that L-arginine-modified nanopolymers are able to increase the biological effect of HBPL towards the original and Omicron strains of Sars-CoV-2. The reason behind this behavior should be attributed to the differences in the way the nanoparticles interact with the virus as competitors to cellular receptors depending on the differences in the structural composition and surface charge. Further studies would be needed to verify this theory and understand the resistance displayed by the Delta variant.

The ELISA results obtained from the post-treatment method (Figure 10d–f) generally show that the antiviral efficacy of the nanopolymers was strongly reduced when the particles were incubated with the cells after viral infection. This indicates that the main effect of the nanopolymers was to reduce the viral replication by inhibiting the virus entry into the cells. This result is in agreement with previous findings on poly-L-lysine hyperbranched nanoparticles. Time-of-addiction experiments have in fact shown a strong reduction in the antiviral activity when the particles were added post infection with respect to a pre-infection incubation [10]. 

The efficacy of the different activities of the nanosystems should, however, be evaluated by also taking in consideration the cytotoxicity data. The antiviral efficacy data taken from the prevention model were used together with the cytotoxicity data to generate a therapeutic index value: TI = CC_50_/IC_50_,(1)

CC = 50% cytotoxic concentration and IC_50_ = 50% inhibitory concentration, i.e., a measure of the balance between the compound’s efficacy and safety. The TI values for the tested nanopolymers are listed in Table 1.

While all the nanopolymers exhibited a low TI when tested with the Delta variant, the original and Omicron strain samples exhibited the highest TI values (high efficacy, low cytotoxicity) with the Lys-Arg copolymers containing the least amount of L-arginine (LBA0.1 and LBA0.25). It is finally worth underlining that, in comparison to the hyperbranched nano-homopolymer, HBPL, the addition of a small amount of L-arginine increased the therapeutic index and the antiviral activity of the resulting copolymers by twelve-fold, while the addition of bigger amounts was accompanied by an increase in the cytotoxic effect. Based on these results, we assume that L-arginine-modified nanopolymers might be suitable as preventive tools to be used in a hypothetical nasal spray device to hinder the infection of Sars-CoV-2 and its variants [28]. Further studies will be needed to evaluate the best formulation for in vivo experiments and exploit the spectrum of activity.

## 4. Conclusions

Hyperbranched nanopolymers obtained via the thermal amidation of L-lysine can be modified by including specific amounts of L-arginine in the precursor mixture and by using boric acid as a catalyst. The nanopolymers have a hyperbranched structure, and increasing amounts of L-arginine favour the formation of more interconnected structures. Larger amounts of L-arginine, however, reduce the solubility, increase the particle size, and reduce the stability of the nanoparticles. The ability to inhibit viral replication was tested on different SARS-CoV-2 strains, such as the original, Delta, and Omicron strains. The evaluation of the therapeutic index gave a direct indication of the correlation between the efficacy and cytotoxicity of the different polymeric compositions to inhibit or block the viral infection in a prevention and post-infection model. Hyperbranched poly-L-lysine modified with a small amount of L-arginine shows a high therapeutic index towards the original strain, with greater than twelve-fold improvements. The L-arginine-modified hyperbranched nanopolymers can also inhibit the replication of the Omicron strain, while the pure hyperbranched poly-L-lysine has a five-times lower efficacy. On the other hand, all the tested nanopolymers show a much-reduced efficacy for the Delta variant. The lack of non-linear trends in the antiviral efficacy as a function of the L-lysine/L-arginine ratio can be explained by considered that both the reduced cell viability and increased antiviral activity contributed to the ELISA outcome. The antiviral efficacy of the nanopolymers was remarkably reduced when the particles were incubated with the cells after viral infection in the post-infection model. In agreement with previous findings, this suggests that the main effect of the nanopolymers is to hamper the viral replication, likely through electrostatic interactions with the cellular receptors or the virions, thus inhibiting the viral infection. The findings of this study open the way for further investigations to confirm the antiviral activity of the hyperbranched nanopolymers for in vivo biological systems and for obtaining a better understanding of the mechanisms of action involved.

## Figures and Tables

**Figure 1 nanomaterials-13-03090-f001:**
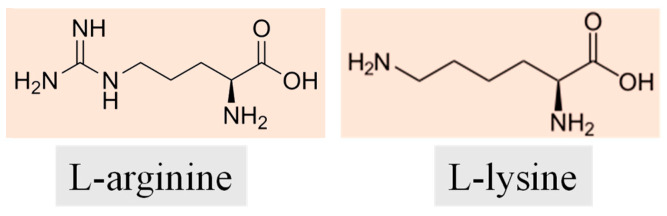
Neutral structures of the amino acids L-arginine (**left**) and L-lysine (**right**) used in the syntheses.

**Figure 2 nanomaterials-13-03090-f002:**
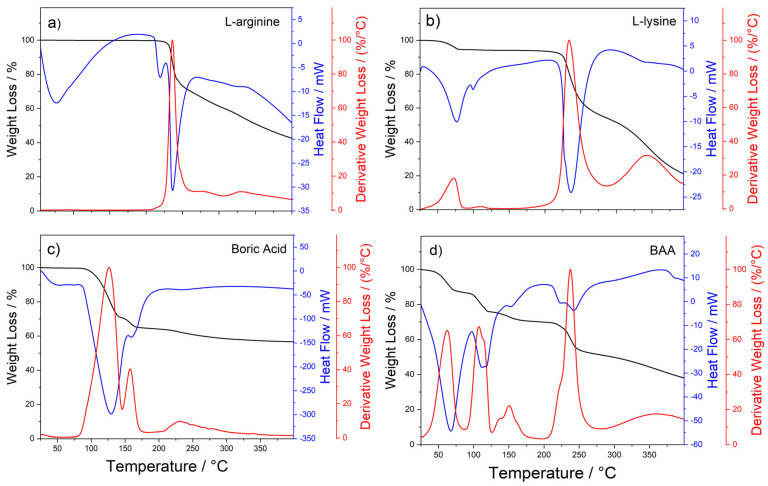
TGA (black curves) and DSC (blue curves) analysis of L-arginine (**a**), L-lysine (**b**), boric acid (**c**), and the 1:1 mixture of boric acid and L-arginine (BAA) (**d**). The first-derivative curve of TGA is shown in red.

**Figure 3 nanomaterials-13-03090-f003:**
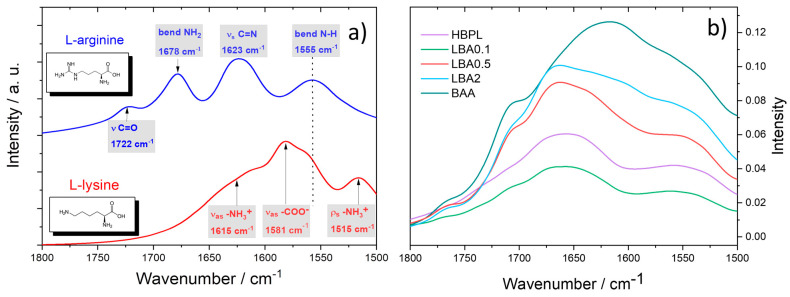
(**a**) FTIR absorption spectra of L-lysine (red line) and L-arginine (blue line) in the 1800–1500 range. The dot line in the figure is a visual guide. (**b**) FTIR absorption spectra of nanopolymers with increasing amounts of L-arginine in the syntheses.

**Figure 4 nanomaterials-13-03090-f004:**
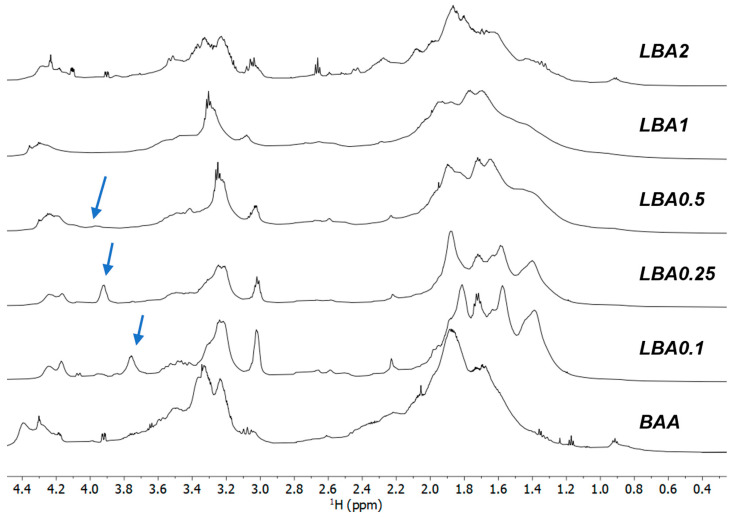
^1^H-NMR spectra of the Lys-Arg samples prepared with increasing amounts of L-arginine. The blue arrows highlight the changes in the α-CH group involved in the amide bonding. The spectrum of BAA is reported as reference. The blue arrows show the progressive growth and shift of the α-CH_2_ groups next to the amide bonds of L-lysine.

**Figure 5 nanomaterials-13-03090-f005:**
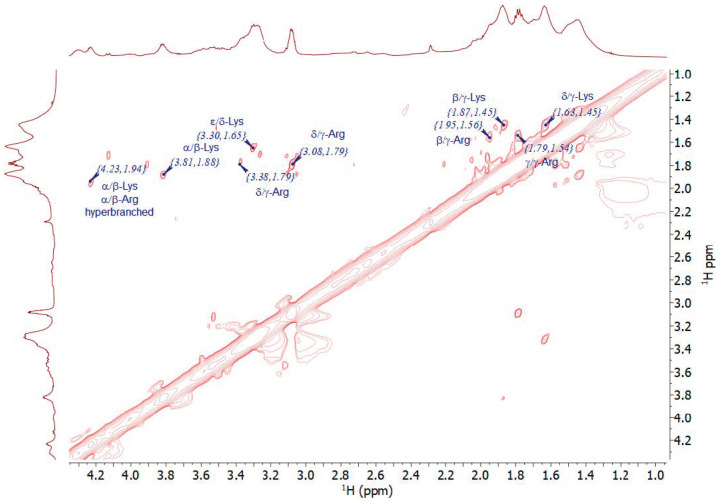
Two-dimensional NMR spectra of LBA0.1 showing the homonuclear correlation (^1^H-^1^H COSY).

**Figure 6 nanomaterials-13-03090-f006:**
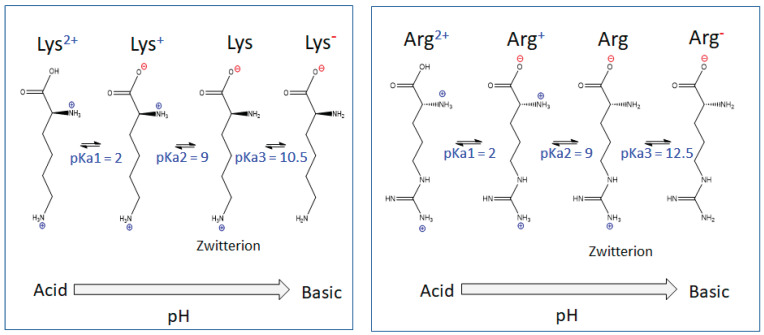
The different protonation states and pKa of the L-lysine (**left**) and L-arginine (**right**) amino acids.

**Figure 7 nanomaterials-13-03090-f007:**
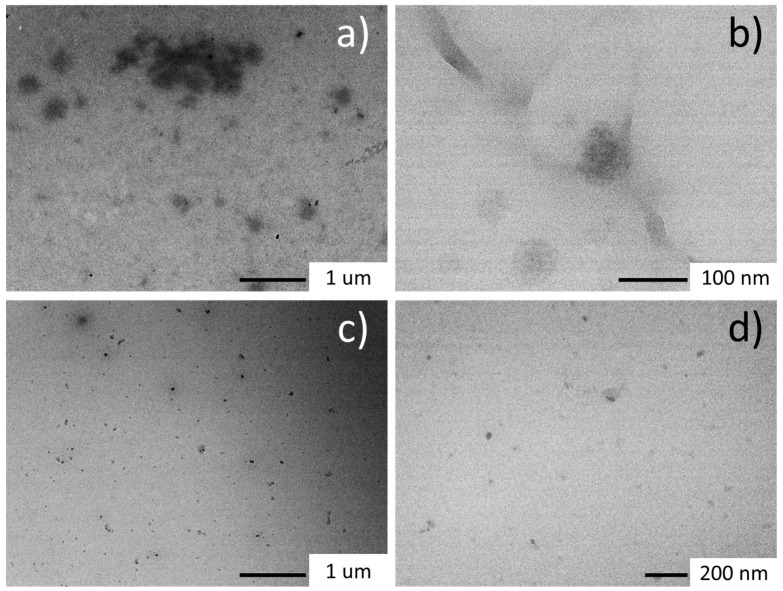
Bright-field TEM images of the nanopolymers at different L-lysine/L-arginine molar ratios. (**a**) LBA0.1; (**b**) LBA0.5; (**c**,**d**) LBA0.25.

**Figure 8 nanomaterials-13-03090-f008:**
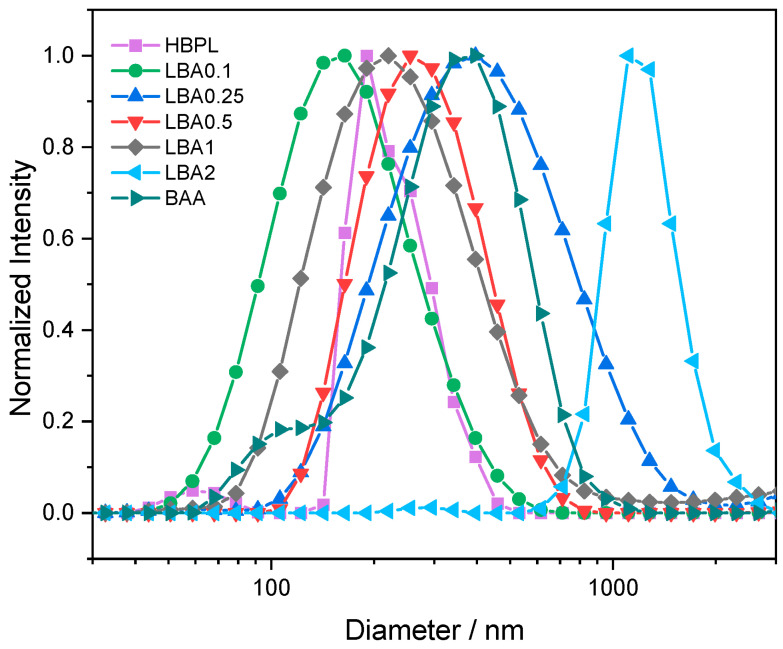
Normalized intensity of the scattering data collected from DLS as a function of the estimated nanopolymer diameter. The lines are a visual guide. Purple squares (HBPL); green dots (LBA0.1); blue triangles (LBA0.25); red inverted triangles (LBA0.5); black diamonds (LBA1); light blue rotated triangles (LBA2); dark-cyan rotated triangles (BAA).

**Figure 9 nanomaterials-13-03090-f009:**
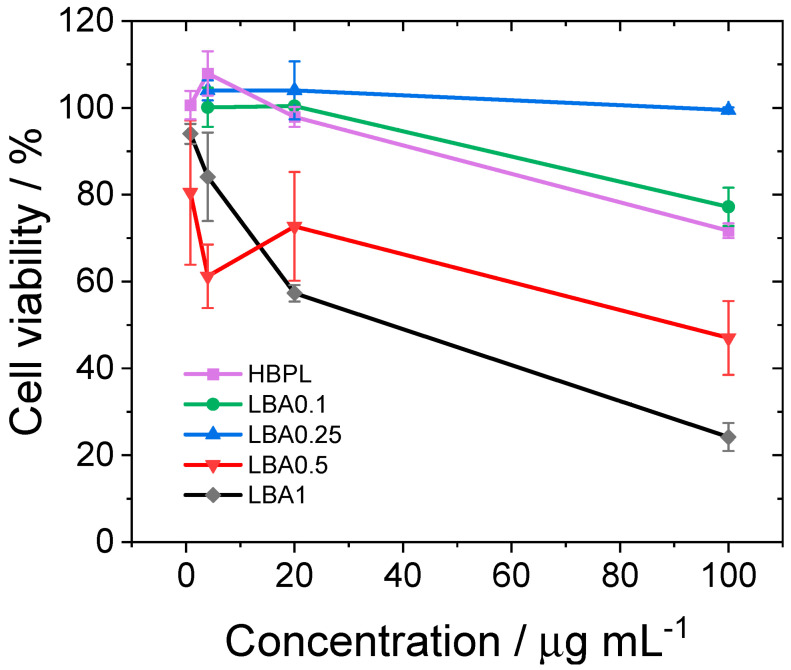
Cytotoxicity of the nanopolymers. The lines are a visual guide.

**Figure 10 nanomaterials-13-03090-f010:**
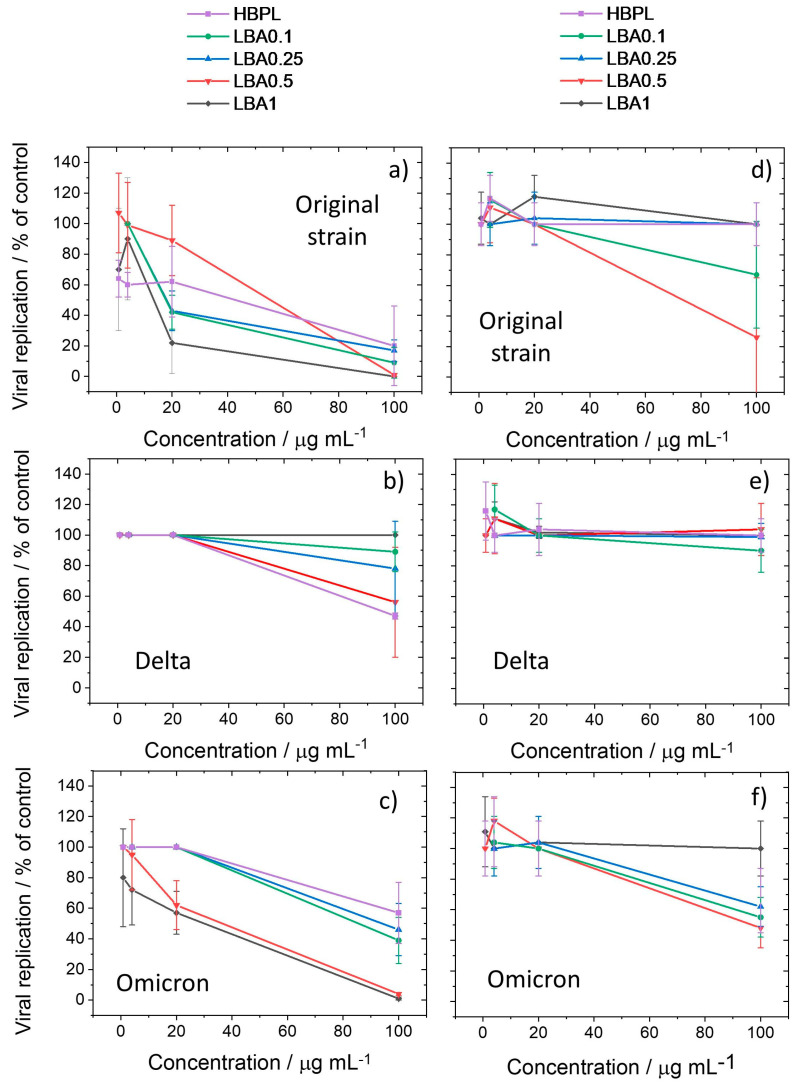
Antiviral efficacy evaluated by the ELISA assay. The assay was used to verify the antiviral properties of the different nanopolymers with respect to the original, Delta, and Omicron strains in prevention (**a**–**c**) and post-infection (**d**–**f**) models.

**Table 1 nanomaterials-13-03090-t001:** Therapeutic index calculated by the CC_50_/IC_50_ ratio.

Sample		Original Strain		Delta		Omicron	
	CC50 (µg mL^−1^)	IC50 (µg mL^−1^)	TI (CC50/IC50)	IC50 (µg mL^−1^)	TI (CC50/IC50)	IC50 (µg mL^−1^)	TI (CC50/IC50)
LBA1	25	13	1.9	100	0.3	25	1.0
LBA0.5	100	55	1.8	70	1.4	36	2.8
LBA0.25	>500	18	≥27.8	≥500	1.0	100	5.0
LBA0.1	>500	18	≥27.8	≥500	1.0	100	5.0
HBPL	100	43	2.3	100	1.0	100	1.0

## Data Availability

All the data presented in this study are available within the manuscript and in the Appendix A.

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
