# Peer review of "In Vitro Antiviral Activity of Hyperbranched Poly-L-Lysine Modified by L-Arginine against Different SARS-CoV-2 Variants"

_nanomaterials, 2023, doi:10.3390/nano13243090_

Round 1

Reviewer 1 Report (Previous Reviewer 2)

Comments and Suggestions for Authors

Dear Authors,

Thank you so much for your revised version, which is acceptable, thanks.

Yin, PhD

Comments on the Quality of English Language

No

Reviewer 2 Report (Previous Reviewer 1)

Comments and Suggestions for Authors

The authors have revised the article or given some reasonable explain according to the opinions of the reviewers, and the quality of the article has been greatly improved. I recommend the manuscript for publication in  Nanomaterials.

Reviewer 3 Report (New Reviewer)

Comments and Suggestions for Authors

The paper entitled  "In vitro antiviral activity of hyperbranched poly-L-lysine modified by L-arginine against different SARS-CoV-2 variants" by Federico Fiori and co. presented the use of the n poly-L-lysine hyperbranched nanoparticles to inhibit the viral replication of different SARS- 87 CoV-2 variants, namely Delta and Omicron in a prevention model by modifying the composition of the hyperbranched poly-L-lysine nanopolymers by co-reacting L-lysine  with another amino, acid, L-arginine. L-arginine is an amino acid characterized by the presence of a guanidinium group appended to an amino acid framework.

The paper is well written and contains original important scientific data, presented and explained very well.

Comments on the Quality of English Language

English is ok

This manuscript is a resubmission of an earlier submission. The following is a list of the peer review reports and author responses from that submission.

Round 1

Reviewer 1 Report

Comments and Suggestions for Authors

In this manuscript,  L-Arginine modified nanopolymers have been prepared and characterized, which proved a good  ability to block both   original strain and omicron strain of SARS-CoV-2. The results are very interesting. On my opinion, I recommend the manuscript for publication in Nanomaterials after some modifications.

1. The TEM images of as-prepared nanopolymers should be provided;

2. In Figure  6 and Figure 7, the error bars should be provided;

3. Indirect immunofluorescence analysis of virus infected cells before and after the addition of L-Arginine modified nanopolymers needs to be providedï¼›

4. Western blot analysis of the expression levels of virus before and after treatment of L-Arginine modified nanopolymers should be provided. 

Comments on the Quality of English Language

No comments.

Reviewer 2 Report

Comments and Suggestions for Authors

Dear Editor,

Thank you so much for inviting me to review the manuscript written by Dr Fiori et al entitled “Antiviral activity of hyperbranched poly-L-lysine modified by L-arginine tested against different SARS-CoV-2 variants”. Authors aim to search for broad spectrum antivirals that can inhibit the infectious capacity of SARS-CoV-2. In their work, it was found that hyperbranched L-lysine nanopolymers have an excellent ability to block the original strain of SARS-CoV-2 infection, have been modified with L-arginine. A thermal reaction at 240°C catalyzed by boric acid yielded Lys-Arg hyperbranched nanopolymers. They also found that the ability of these nanopolymers to inhibit viral replication has been assessed for the original, Delta, and Omicron strains of SARS-CoV-2 together with their cytotoxicity. They also found that the therapeutic index has provided a good indicator of the safety profile and effectiveness of the various polymeric compositions in inhibiting or blocking viral infection. When tested with the original strain, they also found that hyperbranched L-arginine-modified nanopolymers have a twelve-fold higher therapeutic index. They also found that the nanopolymers can also effectively limit the replication of the Omicron strain in cell culture.

The structure and design of the manuscript are acceptable. The language of the manuscript is acceptable. I have some questions to authors before they can be accepted.

Question 1: For materials and methods part, please add catalog number and location of producers for all producers.

Question 2: Line 113-114, what is this figure? Is it a formal figure? If yes, please add figure legend and description.

Question 3: For antiviral activity. Authors mentioned that treated cells with drugs pre-infect viruses. Have you considered to try infection of virus first, then treated infected cells with drugs. Because this is therapeutic model. Pre-treat cells then infect cells is a prevent model.

Question 4: I understand that authors use ELISA to detect viral protein levels, could you also measure viral RNA via qRT-PCR to check effects of drugs on viral replications? 

Comments on the Quality of English Language

None

Reviewer 3 Report

Comments and Suggestions for Authors

My comments are attached.

Comments on the Quality of English Language

The minor english language edition is needed.